# Methodological Proposal for the Analysis of Urban Mobility Using Wi-Fi Data and Artificial Intelligence Techniques: The Case of Palma

Pau Salas [1], Vicente Ramos [2], Maurici Ruiz-Pérez [3] and Bartomeu Alorda-Ladaria [1,*]

1 SmartDestination Research Group, Industrial Engineering and Construction Department, University of the Balearic Islands, 07122 Palma de Mallorca, Spain
2 SmartDestination Research Group, Applied Economics Department, University of the Balearic Islands, 07122 Palma de Mallorca, Spain
3 SmartDestination Research Group, Geography Department, University of the Balearic Islands, 07122 Palma de Mallorca, Spain
* Correspondence: tomeu.alorda@uib.es

**Abstract:** Knowing and modeling mobility in smart city spaces is important for both planning and managing city resources. The optimization of public resources and the improvement of their management are some of the main concerns in the development of sustainable urban development policies. This study proposes the application of several artificial intelligence methodologies to support mobility planning based on data provided by public Wi-Fi infrastructures in the city. Considering that Wi-Fi networks provide high-frequency data about the devices under their coverage radius, three classification techniques are proposed: by frequency of occurrence of the devices, by estimation of the mode of transport, and by estimation of the most common travel routes. As a case study, the city of Palma (Mallorca, Spain), an international tourist destination where mobility is of singular importance, is selected. This study shows the results obtained from a Wi-Fi network with wide coverage that is integrated into the urban space. It provides novel and updatable information on the mobility model of the city by taking advantage of public high-frequency monitoring resources.

**Keywords:** artificial intelligence; real-time urban monitoring; Wi-Fi network data; travel routes estimation; urban modal mobility estimation





## 1. Introduction

It is estimated that the planet has more than 8 billion people, and 56% live concentrated in cities [1]. In addition, it is estimated that in 2050, the population will have increased by 72%, and 67% of the total will live in cities [2]. Given this population growth, we need to adapt the design of cities and their resources to guarantee societies' sustainable future. In this context, the smart cities paradigm has been proposed as a solution to this challenge [3,4] through an intensive and integrated use of information and communication technologies (ICT).

Mobility planning is one of the main challenges derived from the concentration of the population in cities to mitigate the main externalities generated by transport [5,6], which included:

- Atmospheric pollution from emissions.
- Noise pollution.
- Congestion of the main roads due to a high volume of private vehicle traffic, especially due to the dependence of individual transport rather than collective transport.
- Need for high-cost territorial and environmental infrastructures.

In recent years, the mobility as a service (MaaS) conceptualization has received extensive attention [7]. The essence of this approach is that all means of transport are unified to

produce an environmentally friendly mobility model, lowering the cost of maintenance, and optimizing travel times. One of the main shortcomings of implementing such advanced mobility planning is the lack of dynamic information about movements throughout the cities. In general, the applications are only based on static models obtained through mobility studies carried out at a specific moment. Therefore, they are not able to identify trends. By contrast, the planning efforts would be more appropriate if they were based on a realistic mobility model obtained through automatic monitoring systems. This will improve the adaptability of the transport policies of a smart city in the planning stage through the integration of observed trends. But also, it would then be possible to monitor the impacts of the implemented mobility policies in a dynamic context.

Many early urban mobility studies were based on observation and direct counting techniques. However, the data revolution that emerged during the last two decades allows for much more advanced studies. In particular, the proliferation of digital portable devices generated new connection data that was available to telecommunications companies or to the administration itself [8]. This data revolution promoted novel, large-scale mobility studies in cities and at the national level. However, these data sources are not free of limitations, particularly if they do not have continuity over time or show loss of service due to the lack of control the planners have over the data.

Two main methodologies dominate the new information sources based on tracking devices, which can be used to monitor urban mobility, including multimodal options such as pedestrian or bicycle. First, the early applications of Bluetooth sensors for vehicle mobility have also been implemented to model intermodal options such as in [1,2]; second, Wi-Fi network data has also been proposed and implemented [5]. Any of these applications, was usually limited by the absence of a unified network covering a significant section of the urban space [7].

In this context, this paper aims at presenting a dynamic urban mobility study that covers major parts of a city. Our proposal is to use a city's public Wi-Fi network to automatically monitor real-time mobility through detecting and analyzing the devices that are within the coverage area of these infrastructures. The use of high-frequency data can be particularly informative in socioeconomic studies, as it is common that these data present structural beaks [9] that might affect the results obtained with lower data frequencies. Our methodology takes the overall network information and disaggregates it into smaller subsets of knowledge that can be used to inform different urban planning policies. Section 4 presents different typologies of data analysis aimed at characterizing the types of devices captured by public Wi-Fi networks and evaluating the mobility patterns for non-fixed devices. Specifically, we propose the following applications:

- First, there are some devices that appear regularly in the database, while others only appear a few times. In terms of mobility monitoring and planning, it is relevant to understand the behavior of these two groups. Therefore, we classify the devices according to their appearance on the network. In this sense, we implement a protocol to distinguish between devices that appear often (usual) and those that appear a few times (sporadic). Additionally, for the usual devices, we evaluate if they move (dynamic) or stay at the same location (fixed) within the coverage area (spatial dimension) and throughout the monitoring period (temporal dimension).
- Second, one advantage of the Wi-Fi network monitoring is the possibility to capture multimodal transport options. To inform urban mobility policies, we estimate the mobility mode of usual dynamic devices.
- Finally, one of the main advantages of covering a huge area of the city is that we can model spatiotemporal patterns. To do so, we detect the most common routes and the most visited singular spaces of the city.

Regarding data privacy and the protection of the user's identity, the technical Wi-Fi data only use the device's MAC address. According to the European Regulation ([8]), this is not considered personal information as it does not allow users' identification. The data do not include any record or information regarding the device's owner. The data

come from a public and free-of-charge network that does not have any user records. It is worth mentioning that our methodological proposal does not require to: install any type of software on users' devices; collect data that can be associated with an electronic identity; or impose any restriction on the infrastructure's discovery mechanism of any operating system or mobile terminal [10,11]. In fact, this is one of the main contributions of our proposal. There are few previous studies [10,12] that use complete urban public Wi-Fi to monitor the real-time urban mobility of all devices that move throughout a city. This infrastructure is not oriented to the location of devices and does not impose any restrictions or conditions on including a device in the study.

In the following section, we present the main characteristics of studies that use private Wi-Fi infrastructures. Section 3 describes the public Wi-Fi network that generates the raw data. Next, Section 4 details the different methodologies, while Section 5 presents the data analysis. Finally, Section 6 summarizes the main conclusions of our research.

## 2. State of the Art

A new type of data emerges from tracking portable digital devices in the era of the Internet of Things. These are characterized by a huge number of observations with detailed spatiotemporal information. A very important task in managing all this information is the automatic classification of devices detected through a communications infrastructure. Along these lines, Bai [13] classifies the devices with deep learning algorithms based on the use of neural networks. In the identification process, the traffic is differentiated according to the protocol used (user or control) and the volume of transferred information. The learning phase adjusts the predictive capacity, which, in the case of [13], reaches 74.8%. This methodology requires recording the devices' traffic to define the neural network parameters. Thus, to a certain extent, the confidentiality of the data is violated.

To avoid using devices' traffic data, the authors of [9] analyzed the devices' behavior using machine learning algorithms. The final objective of their research was to classify the devices in order to implement control measures that reduce mobile phone network maintenance costs. The authors applied different regressions and statistical algorithms and used a "random forest" approach that produced a successful prediction rate of 79.29%. One characteristic of this method is that it requires previous knowledge of the users' transport mode. Additionally, it requires the estimation of a large number of parameters. Due to these two requirements, this approach is not appropriate to model data from a public Wi-Fi network, where the modes of transport of different devices are not known with certainty.

The objective of paper [7] is very similar to our proposal. They also estimated the mobility mode (foot, bicycle, and car) using Wi-Fi signals and proved that these data could be used for this purpose. However, their application proposes a scenario in a closed circuit in which the modes of transport of each device are known. Machine learning algorithms are used to determine the modes of transport, and as in the previous case, a database of the characteristic parameters of each mode is required. These requirements cannot be applied in an open and real scenario since each device can vary in travel mode along its route.

Additionally, the estimation of the most common routes has also been explored to improve communications' efficiency in delay-tolerant networks (DTNs). Study [14] determines the city locations and times at which there are high device concentrations. One relevant application of this method is to identify the optimal devices for data dissemination, as they will coincide with many other devices with which they communicate. Authors in [14] found that an analysis based on historic data of each device's geolocations during the previous days may improve the efficiency of the most visited city location results by 183%. In this sense, a methodology to determine the most common mobility patterns in a city may be essential for smart city mobility managers.

Finally, Table 1 shows a collection of previous works related to the use of Wi-Fi networks in urban mobility applications.

**Table 1.** Includes a benchmark analysis of the use of Wi-Fi networks in urban mobility analysis.

| Reference | Objective | Location |
|---|---|---|
| [15] | Analysis of pedestrian network patterns and their interactions. | Hong Kong |
| [16] | Monitoring of passenger flow in public transport. Generation of origin–destination matrices from Wi-Fi data. | King County, Washington (USA) |
| [17] | Multimodal public transport route choice. | Copenhagen (Denmark) |
| [18] | Wi-Fi in transport network traffic evaluation and generation of O–D matrices. | Surat (India) |
| [19] | Analysis of people density, flows, dwell times, return times, heat maps. | Caliari (Italy) |
| [20] | Route choice and travel time estimation in tourist areas. | Kyoto (Japan) |
| [21] | Spatio-temporal route detection in rail transport. | Shanghai (China) |
| [22] | Evaluation of pedestrian routes in urban environments. Detection of utility walks and leisure walks. | London, UK |
| [23] | Bus route demand assessment. | Shanghai, China |
| [24] | Analysis of passenger volume and origin–destination matrices of a bus route. | Obuse, Japan |

## 3. Data Sources

### 3.1. Case Study: Public Wi-Fi in Palma

In 2015, Palma City Council began a large-scale project called Smart Wi-Fi Palma, with the advice of the University of the Balearic Islands (UIB). This project consists of installing a metropolitan Wi-Fi network formed by a set of access points (APs) that allow wireless interconnection with mobile devices. These APs covered the main city center and tourist areas. Figure 1 shows different pictures to locate the study area, with indications about the APs installed in the city (green dots) and their coverage area (red shadow). This study uses the data from the summer months of 2019 (between June and September). It should be emphasized that the Balearic Islands are one of the major tourism destinations in the Mediterranean. In 2019, more than 16 million tourists visited the region [25]. Moreover, tourism demand is highly seasonal, with more than 62% of tourists arriving between June and September [26]. Therefore, July and August are the months in which there is higher urban pressure in the region's capital, Palma.

All mobile devices that have their Wi-Fi transmitters activated and that are within the coverage area of the smart Wi-Fi can be monitored by the network.

### 3.2. Monitoring Architecture

The installation of the APs throughout a city is not sufficient to provide a monitoring system. Therefore, the smart Wi-Fi network is implemented with Cisco® Meraki® technology. This tool processes anonymously the devices' download packets and uses this information to estimate their geographical position within the network coverage area (see [12,27] for further details). Additionally, it has an application programming interface (API) that provides the set of events identified by each AP, collecting them in JavaScript Object Notation (JSON) format. The overall process is shown in Figure 2, where the Meraki® cloud acts as a raw data source in JSON format. These data are received at a UIB server and downloaded into a PostgreSQL database for further processing and automatic analysis.

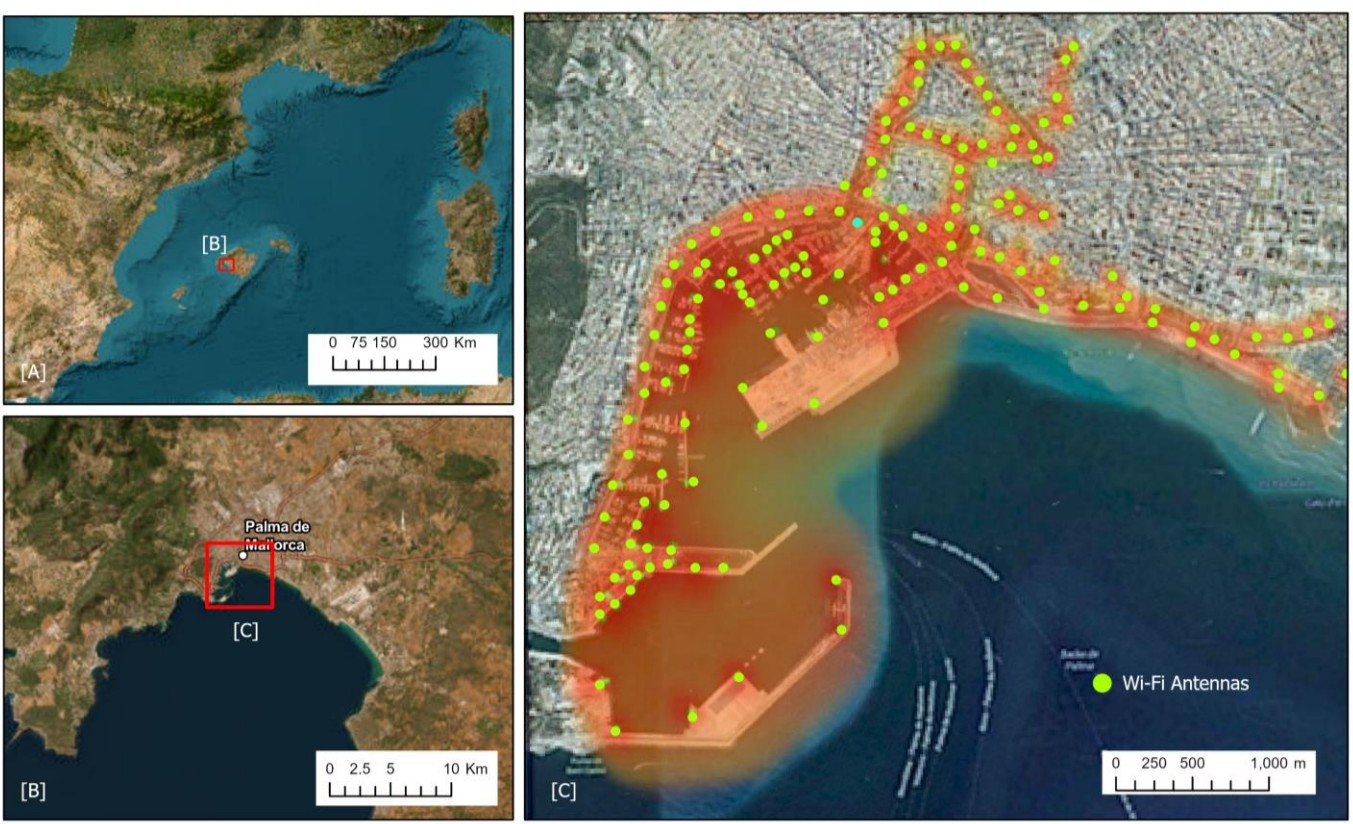

**Figure 1.** (**A**) Location of Palma at Mediterranean basin; (**B**) location of study zone at Mallorca Island; (**C**) location of study area with APs distribution and the Wi-Fi coverage.

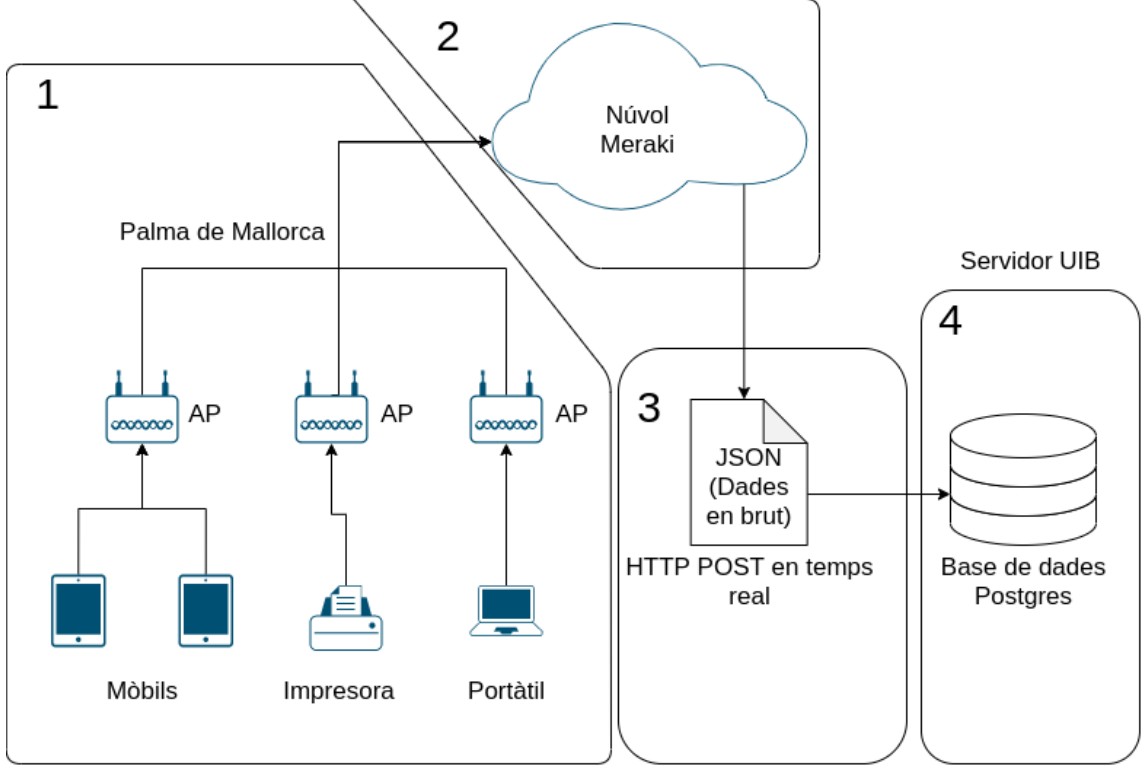

**Figure 2.** Capture processes of the downloaded packets in the same devices.

The Postgres database stores the variables summarized in Table 2 that are associated with each device communication treated by the location system network.

**Table 2.** Summary of the general Wi-Fi location parameters of the smart Wi-Fi network received in the JSON file.

| Parameter | Description |
| --- | --- |
| APmac | MAC address of the first AP that receives the device's discovery packet. |
| Seentime | Device's detection time stamp in epoch format. |
| Lat | Estimation of the device's latitude coordinates. |
| Lng | Estimation of the device's longitude coordinates. |
| Clientmac | MAC address of the device that generates the discovery packet |
| Model | Name of the device manufacturer obtained from the initial 24 bits of the device's MAC address. |
| SSID | If the device is linked to the network, this is the name of the SSID used. If there is no link, this field is empty. |

The network generates a massive amount of data that arrives in real time to the server. Therefore, data recording is subject to technical challenges that need to be considered when applying the proposed methodology. In the next section, we discuss some considerations for managing and filtering the data to get a stable, complete, and robust database.

*3.3. Data Cleaning*

Automated data monitoring systems should incorporate information cleaning mechanisms to maintain consistency in time series information [28]. There are several events that might lead to information degradation:

- The devices' detection, the generation of its associated information, and its arrival to the server are not instantaneous. In other words, data processing, transmission, and storage take varying amounts of time, which must be considered.
- The semantics of the information fields are weak. In some cases, there are no data, or the data are not well defined.
- Some parameters' values might be inaccurate. Some examples include deficiencies in the geographical location estimation, or loss of the event's temporal reference.

Therefore, it is necessary to incorporate information-refining processes that provide consistent and adequate data for the application of the proposed methodologies. In our case, the criteria that have been applied to filter the raw data include:

- The detections happen at different coverage areas throughout the city. Therefore, its arrival to the server does not necessarily follow the events' chronology. Data require subsequent temporal ordering through the "*seentime*" parameter.
- The geographical location latitudes might occasionally be imprecise. This estimation error is random and difficult to process. In our database, the records located outside the coverage area are eliminated to make the information more consistent.
- Some entries show a device in two different places at the same instant of time. In this case, the debugging process consists of eliminating the affected observations.
- The speed estimation in some entries was above 80 km/h. These speeds are not possible in a city with maximum speeds of 50 km/h for vehicles, especially when they are obtained in the city center and pedestrian areas. For this reason, this information is replaced by speed extrapolated from previous and subsequent periods for the same device.

## 4. Multimodal Urban Mobility Monitoring Methodologies

This section describes the statistical methodologies that we used to analyze the smart Wi-Fi data. Its application illustrates that public Wi-Fi infrastructures can be used to improve urban mobility model knowledge. Specifically, we implemented three mobility monitoring applications, which can contribute to decision making by city planners and managers.

### 4.1. Classification of Devices

We start by acknowledging that the capturing process will detect any digital device that appears in the coverage area. However, we can extract different useful information from specific devices' typologies. More specifically, the dataset includes devices that appear sporadically and other devices that are often captured by the network. Depending on the research purpose, it is relevant to distinguish between these two groups. For example, crowdedness studies require considering all devices that are detected at a given point in space and time; however, route analysis should be implemented on devices for which there is a higher temporal footprint on the dataset.

Therefore, two types of devices are identified:

- Sporadic devices: devices that appear in a few events during the day.
- Frequent devices: devices that appear in many events.

Hence, we will label as sporadic, those devices that have been briefly in the network's coverage area. Several reasons can explain why the dataset includes devices with very few observations. The most likely are those cases in which a device moves within the limits of the network's coverage in any slow mobility mode (foot, bicycle) or crosses through the city in a motorized vehicle (public or private). Figure 1 indicates that the network covers a vast area of the city. Hence, the area's perimeter is very long, and it also includes crowded motorized routes that cross the city. It might also be the case that some individuals decide to switch off the device in case it runs out of battery.

Due to data variability, the devices' classification as sporadic or frequent is more complex than it might seem. For example, there were some devices that had more than 20 detections in only 5 s, and they do not appear again. Hence, to calibrate a threshold for the classification, we propose to consider both the number of times that a device is detected and the time gap between those events. In our database, we estimated that the average number of unique devices per day was 21,365 devices, with an average of 3.57 observations per device. For any given day, the device with more observations was captured 350 times, and the mean time between observations was approximately 2 min.

This calibration proposal facilitates the adaptation to the possible particularities of each Wi-Fi infrastructure in further applications of the methodology. Additionally, the thresholds could be defined by zone if there are specific peculiarities for different areas. In our empirical application, the Wi-Fi network covers areas for which there is some uniformity in the recording of events. Hence, we used the same calibration throughout the city.

Additionally, we acknowledge that the network identification process can identify devices that do not correspond to individuals moving across the urban space. Therefore, we proposed a second characterization for frequent devices based on their geographical mobility. In this sense, we distinguish between:

- Fixed devices: always appear at the same location, and are detected by a single AP.
- Dynamic devices: are registered by several APs and move through different locations.

Therefore, the proposed classification segments the devices into the following groups: sporadic, frequent-fixed, and frequent-dynamic.

### 4.2. Estimation of the Mobility Model

As we emphasized in the introduction, one of the main contributions of this research is that we implemented the data analysis on a vast area of the city (Palma, Spain). In addition,

the purpose of the research is to provide usable knowledge that can be incorporated in urban planning. Particularly in the context of tourism, transport mode has often been included as a relevant variable explaining the behavior [29]. Therefore, we also aim at explaining the mobility patterns considering a multimodal model. Those devices that are classified as dynamic and frequent will be captured at different spans in space and time, and it is therefore possible to estimate a speed and extrapolate the mobility mode. We propose a methodology to identify three mobility modes that are relevant for urban planning scenarios:

- Still mode: devices that remain at the same location during a given period. For example, they could be those of people who are in a commercial establishment or inside a restaurant.
- Pedestrian mode: exhibit a low-speed movement through the geographical area covered by the Wi-Fi network.
- Fast mode: all situations in which devices move at a speed faster than that of a pedestrian: running, using a scooter, bicycle, motorcycle, car, or public transport, etc.

The classification of the devices in each of these three categories is based on three variables:

1. Estimated speed. The range of speeds for each modality, shown in Table 3, is defined in [7]. Differentiating the other modes of transport with greater precision requires a process of characterization of each area of the city, as indicated in [7]. In addition, the use of the Wi-Fi network is identified as the most appropriate for the detection of low-speed modes.
2. The number of APs that have detected the device. This is considered since it indicates that the device has moved within the coverage area.
3. Number of areas of the city that have been visited with the same modality. The city is divided into different areas, which are associated with different activities. This zoning groups the APs of the entire network. In the context of characterizing the mobility mode, this variable helps to understand the mobility model.

**Table 3.** Mode associated with speed [7].

| Mode | Speed Range (km/h) |
| --- | --- |
| Still | $\leq 2$ |
| Pedestrian | between 3 and 7 |
| Fast | $\geq 8$ |

However, the identification of the mobility mode of a mobile device in an open space is a rather complex process. Some of the challenges are associated with the behavior of the device. During a given day, the same device can be recorded using various mobility modes. Thus, it is necessary to determine the periods for which the device exhibits a homogeneous mode of transport. Additionally, the state of the device can change (sleep mode, low consumption, or other situations that decrease the frequency of discovery). In other words, the observations are not uniformly distributed over time; therefore, there are periods with no observations, periods with over-observation, and periods of detection at different areas of the city (as they are bordering each other). Other limitations are related to the inherent difficulty of not having prior users' information. It is, therefore, an open scenario in which the different modes of transport in each coverage area have not been previously characterized (as was the case in reference [30] with a closed analysis environment).

We have considered all these situations. Hence, the data are processed to eliminate errors in the estimation of the modality. The mobility mode depuration process includes the following tasks:

- The periods without an appropriate number of observations are detected by establishing a maximum threshold between records. We established this value at the third quartile of the time between observations. Therefore, if the time gap between two detections is above the threshold, they are considered to be part of two different periods, and these two detections are treated independently in the classification of mobility mode.
- The periods of over-observation are probably due to situations or uses in which a device generates discovery packets at a higher frequency than usual. Hence, the distance between both temporal and spatial observations will be very similar. However, this fact does not affect the mobility classification.
- Finally, if a device is located at the boundary of different coverage areas, it might be detected by APs associated with each of them. If this is the case and depending on the distribution of the radiation pattern of the APs, it might produce geopositioning mistakes. This will subsequently lead to errors in the estimated speed or the number and chronological order of the city areas that have been visited. Therefore, these observations have been removed from the data set.

Considering all the above specificities, our methodological approach is to apply a multimodal logistic regression algorithm for the classification of mobility modes. This algorithm uses maximum likelihood estimation to evaluate the probability of allocating each observation to a given category. According to Tabachnick [31], this algorithm has the following advantages over other statistical classification techniques:

- It is more robust as it does not require assuming normal distribution of any variable (or the error terms), or equal variance and covariance matrices among the categories.
- The estimation statistics are easily interpretable.
- It does not assume any type of linear relationship between the independent and dependent variables.
- There are no interval restrictions on the independent variables.

The application of multimodal logistic regression requires the definition of a table of relationships between the independent variables and the dependent variables. In this study, the procedure was as follows:

1.  We first estimate the relevant observation period for each device and mobility mode.
2.  We calculate the probability of each speed range, according to Table 3, following the expression:

$$\frac{number\ of\ mode_i\ observations}{total\ number\ of\ observations\ at\ the\ period} = Prob(mode_i) \tag{1}$$

where "$mode_i$" is one of the three modalities defined in Table 3.

3.  The probabilities of the rest of the independent variables are determined for those bands with a mode probability above 50%

a.  Probability of different APs:

$$\frac{number\ of\ different\ APs}{total\ number\ of\ APs\ at\ the\ period} = Prob(APs) \tag{2}$$

b.  Probability of different city areas:

$$\frac{number\ of\ different\ city\ areas}{total\ number\ of\ city\ aread\ at\ the\ period} = Prob(\text{Areas}) \tag{3}$$

Figure 3a shows the correlations between modes based on speed. We can observe that the "still" mode is not correlated with the other two modes, while the "pedestrian" and "fast" modes may present some difficulties due to the existence of nonzero dependent probabilities.

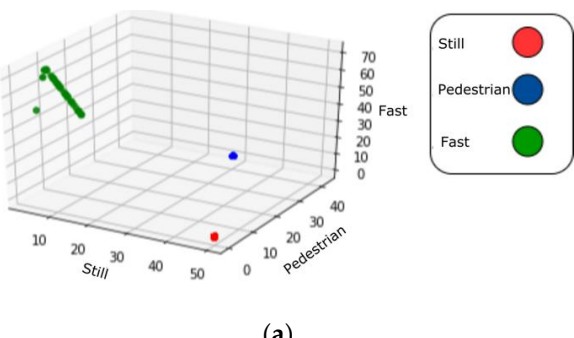 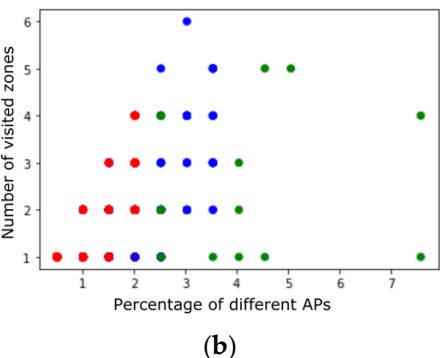

(**a**)                                                             (**b**)

**Figure 3.** Estimation of the correlation between the probabilities of the independent variables. (**a**) Correlation between modalities and speed; (**b**) correlation between modalities and percentage of APs and number of areas visited.

Figure 3b shows the correlations with the dependent variables (number of APs and number of city areas) according to mode of transport. We can see that still mode intersects with visiting multiple areas, but with few APs. The figure also shows the difficulty in unequivocally distinguishing the modalities of "pedestrian" and "fast" when the device presents a low number of observations from APs and different city areas.

Section 5 presents the results of applying this mobility mode estimation method to the city of Palma.

*4.3. Detection of Probable Routes*

The mobility of people in a city is governed by complex socioeconomic models in which the availability of infrastructure and services, together with the preferences of users, play a fundamental role. A route provides connectivity between origin and destination points or zones. We aim to identify the main routes that are used to move around the city using Wi-Fi data. Therefore, we can only study those movements inside the network coverage itself. Additionally, this exercise is also restricted by the characteristics of the network (number and position of antennas, etc.). The process of route detection starts by selecting the origin and destination areas to monitor and, subsequently, identifying the mobility of devices between them. These areas can be areas of interest for the city in terms of commerce, industry, etc. Alternatively, they can be adjusted to the criteria of the planners.

In this study, several areas of interest have been defined using a touristic criterion. In Figure 4, these areas are squared in red and named. Although more areas can be determined, these are the most relevant adjustments to the Wi-Fi coverage area in Palma City.

The methodology that we propose for the analysis of probable routes is based on the application of association rules (AR) theory. This approach is used in data mining to find correlations between datasets. In its application to urban mobility, AR theory identifies the most likely antecedent spaces when a certain observation occurs in a specific area of the city. AR are measured with indicators of their importance to identify the most likely regulations. In our proposal, the indicators that are used to identify the most likely association are:

- Support [32]. Measures the frequency of a rule (movement between $X$ and $Y$) among all possible rules (movements):

$$Support(\{X\} \rightarrow \{Y\}) = \frac{\text{\# movements between } X \text{ and } Y}{\text{\# Total movements}} \quad (4)$$

- Confidence [32]. Measures the frequency of being at $Y$ after being at $X$.

$$Confidence(\{X\} \rightarrow \{Y\}) = \frac{\text{\# movements between } X \text{ and } Y}{\text{\# movements from } X} \quad (5)$$

- Lift [11]. Relates the overall movements between *X* and *Y* and all ones that have {*X*} as an antecedent. A value close to 1 indicates that *X* and *Y* are independent.

$$Lift(\{X\} \to \{Y\}) = \frac{Confidence}{Support} \tag{6}$$

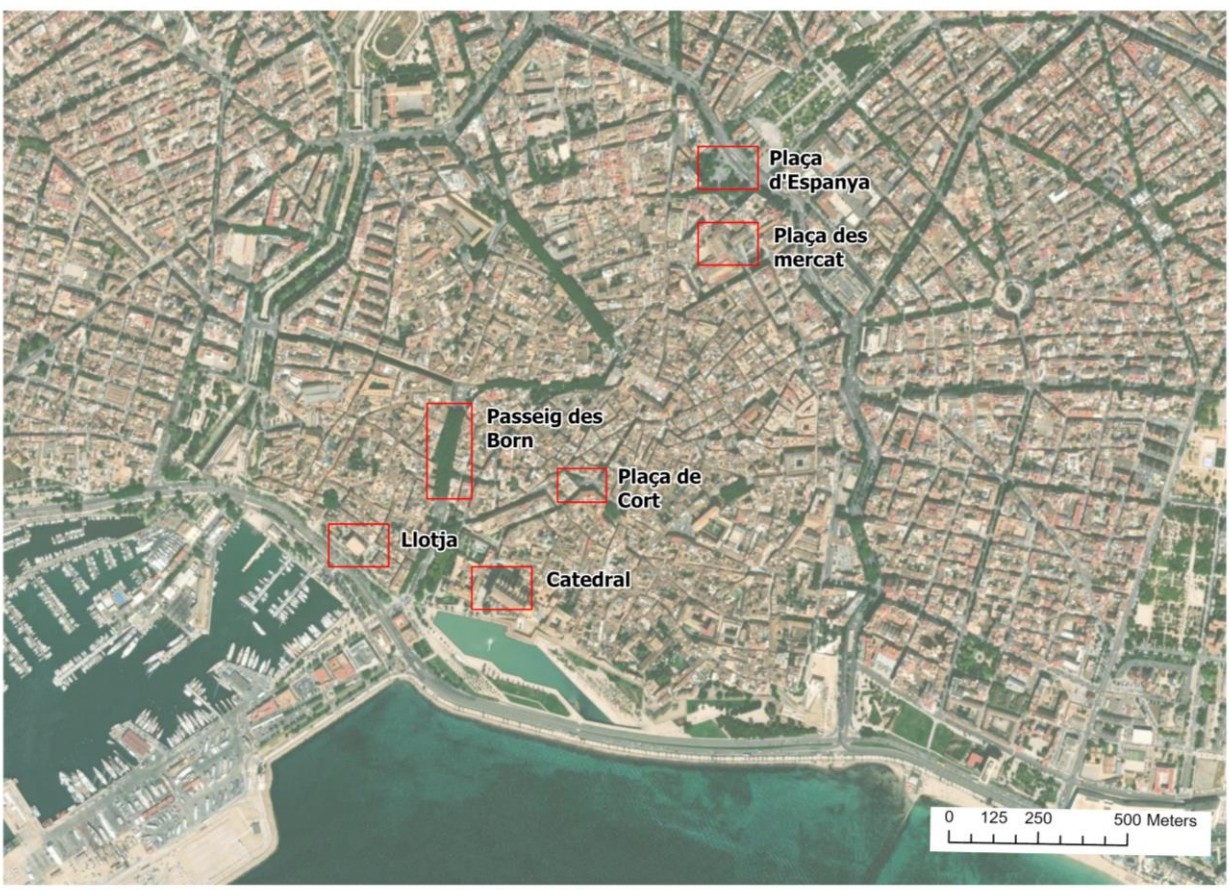

**Figure 4.** Map of Palma City with the interest areas squared in red.

The implementation of the methodology for the estimation of the most likely routes is based on the following task scheme:

1. Define the relevant areas of the city that will be considered.
2. Identify the observations classified as "frequent" and "dynamic" using the methodology that has been defined before.
3. Obtain all possible associations between the areas from the selected observations.
4. Order the association rules by their level of support.
5. Select the association rules from highest to lowest confidence and/or Lift.
6. To conclude: identify the most likely routes among the study areas.

In our empirical application, we divide the overall dataset into three time frames to allow for different routes at different hours during the day.

## 5. Results and Discussion

The public Wi-Fi infrastructure of Palma City covers the city center, which includes the areas of greatest tourist influx. The objective of this infrastructure is to improve internet connectivity for both residents and tourists in the city. However, through this paper, we prove that the network raw technical data can be used to monitor the mobility throughout the city at a very precise temporal and geographical scale. This knowledge can be used to

understand the urban mobility model, which is a relevant piece of information for urban planning and management.

Most previous studies that applied similar technologies implemented them in closed or controlled spaces or in much more limited contexts such as university campuses. Going beyond those limits, towards a bigger, heterogeneous, and multimodal space, introduces relevant research challenges.

In this section, we present the results using data from the first week of July (July 1 to 7) of 2019. It is important to note that this period corresponds to the beginning of the high tourist season of a mature tourist destination.

### 5.1. Classification of Devices

During the study week, the cumulative total number of unique devices detected by the network was 943,839, which implies an average of around 135,000 unique devices per day. When we applied the classification method described in Section 4.1, we obtained the following daily desegregation: approximately 115,000 devices were identified as sporadic, while 24,000 devices were classified as frequent. This last set of devices can be further divided into approximately 23,000 devices identified as dynamic and 1000 categorized as fixed devices. As could be expected, the number of devices is not equal across all the days of the week. Hence, Figure 4 shows the device classification evolution during the week.

The weekly pattern shown in Figure 5 indicates that the flow of sporadic devices is increasing from Monday to Friday and that there is a significant decrease during the weekend. It can also be observed that, as expected, the network captures a huge number of sporadic devices (nearly 80% of the total). Those devices do not spend enough time in the coverage area of the network, or they are detected by some AP at the limit of the network area. Probably that is related with people that just move around the borders of the coverage area or cross the coverage area using a fast mobility modal.

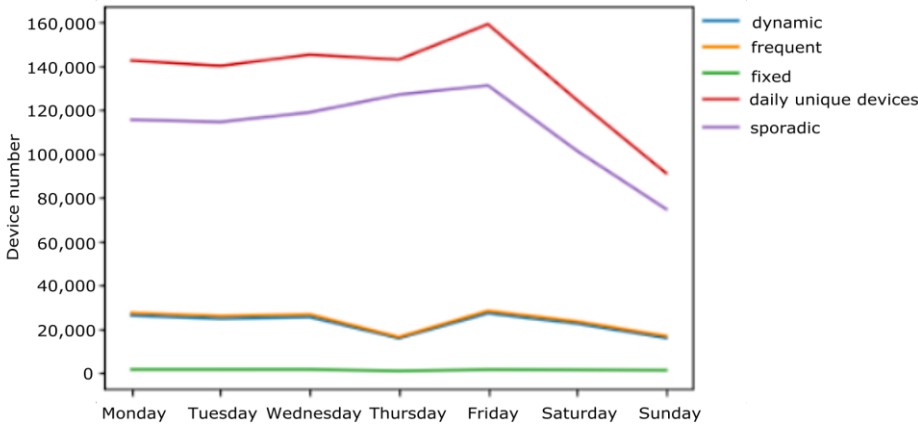

**Figure 5.** Weekly evolution of the device classification.

The frequently used devices represent the remaining 20%. This group can be further classified as fixed or dynamic devices. The former represents around 1% of the total number of unique devices, and its weekly pattern describes a constant and small part of the overall sample, as shown in Figure 5.

Finally, the dynamic devices are those that provide richer information to understand the urban mobility model. During the week under consideration, those frequent devices decrease slowly during the first days of the week, reach a minimum on Thursday, increase on Friday, and then decrease again during the weekend (but at a lower slope than the sporadic devices).

### 5.2. Mobility Mode

The device classification methodology described in the previous section is used to create a subsample that includes the devices identified as frequent and dynamic. This subset is denoted as the most useful for urban mobility modeling and extraction. The multimodal logistic regression method described in Section 4.2 is applied to identify an average of approximately 60,000 daily device trajectories. The maximum daily trajectories are obtained on Friday, while the minimum on Sunday. Figure 6 shows the results of applying the statistical tool to classify the device trajectories depending on the mobility mode. Most of the trajectories are associated with pedestrian mobility throughout the city (with an average of approximately 30,000 trips), followed by fast modes. These results reflect the usual walking related activities that take place in Palma's city center, mainly because we are analyzing a high-tourist season summer week.

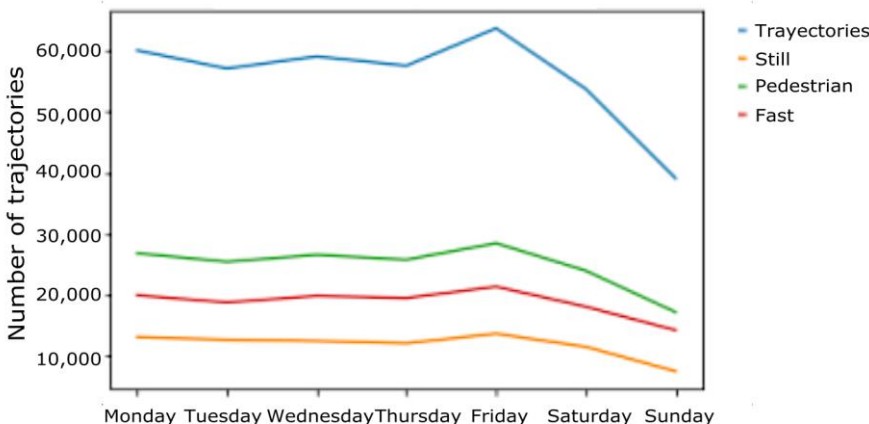

**Figure 6.** Evolution of the number of trips by mode of transport.

### 5.3. Estimation of the Most Common Urban Routes

Finally, this study tries to evaluate the viability of determining the most common routes that are used to move around Palma City during the selected week of July during the high tourist season. This is a crucial piece of information for urban planning in a mature destination that aims to optimize mobility planning. Among many other policy actions, it could be used to: improve the urban infrastructure of those routes; modify the traffic light frequency; increase public resources on those routes; locate public or private services; and design policies to stimulate alternative routes, etc.

In this section, we apply the association rules methodology described in the previous section to characterize the preferred routes between selected areas of the city. This study limited the calculation time by establishing a minimum support value (see Equation (4)) equal to 10% and a minimum lift value (see Equation (6)) equal to 0.1. These values could be adjusted to reduce the number of possible routes considered in the association rules methodology. The limits defined in this study were obtained after some prelaminar evaluation of the whole dataset in order to focus just on the three most preferred rules.

Additionally, although it is not necessary for applying the methodology, some temporal distinction is introduced to provide more useful results regarding the most likely movements through the city. Each daily dataset of 24 h has been divided into three periods:

- Business hours: Monday to Friday between 8:00 and 17:00.
- Entertainment-leisure: Monday to Friday between 17:00 and 00:00 and Saturday and Sunday from 10:00 to 00:00.
- Rest period: Monday to Friday from 00:00 to 8:00 and Saturday and Sunday from 00:00 to 10:00.

The results obtained from the application of the proposed methodology for each time frame are summarized in Tables 4–6. The tables show the routes obtained with the

maximum values of each of the parameters of interest, introduced in Equations (4)–(6) highlighted in black.

**Table 4.** Results for the routes during the work period.

| Rule | Support (%) | | Confidence (%) | | Lift | |
|---|---|---|---|---|---|---|
| | **Monday** | **Tuesday** | **Monday** | **Tuesday** | **Monday** | **Tuesday** |
| {Catedral} to {Passeig del Born} | **50.1** | 45.7 | 89.5 | 87.9 | 1.13 | 2.14 |
| {Plaça del Mercat} to {Passeig del Born} | 45.6 | **48.3** | 93.9 | 92.6 | 1.19 | 1.2 |
| {Llotja de Palma, Catedral, Plaça del Mercat} to {Passeig del Born} | 12.2 | 11.9 | **99.36** | **99.6** | 1.27 | 1.29 |
| {Catedral, Plaça de Santa Eulalia} to {Plaça de Cort} | 10.6 | 10.8 | 84.2 | 86.7 | **2.87** | **3.09** |

During business hours, there are two routes with greater support than the rest, with high values of confidence and lift. Such as, a high support indicates that both routes are the most frequent route. In addition, a high confidence value highlights that the destination is the most frequent considering the origin. Finally, a high value of lift denotes a certain degree of dependence between origin and destination places.

On the other hand, some routes are also characterized by a lower support value but higher confidence or lift values. This indicates that although they appear less frequently in the dataset, the places involved in the rule are strongly related and must be considered.

In this sense, Table 4 shows the four rules selected by the authors of this study in the business hours period. The first and second rows are the most frequent urban routes. The third row is less frequent, but the destination is highly probable if the origins are those included in the rule. It is important to note that the rules in the first and second rows are included in the rule in the third row and extended to a new place. Finally, the rule in the fourth row identifies a highly dependent relation between places although they are not highly probable in the dataset.

A similar selection of urban rules has been performed for the leisure-entertainment period in Table 5 and for the rest of the hours in Table 6.

In the entertainment-leisure time frame, the rules with the highest frequency have a maximum support value lower than those shown in Table 4. This result may be explained because there is an increase in the number of routes due more motives for that route. In other words, during this time slot, people may move around the city with a lot of possible destinations.

In addition, two final rules are included in Table 5. These correspond to routes for which it was not possible to obtain the parameters for all the days, but they obtain the highest results on certain days of the week. The rules in rows 6 and 7 in Table 5 achieve the highest lift values only on Saturday and Sunday, while the rest of the days the values of these rules do not present enough consistency.

Table 6 presents the results of the third group of hours labeled "rest period". The selected rules are explained in a similar way as in Table 5. It is remarkable that the urban mobility randomness is more evident, as indicated by the low levels of the support parameters.

As a summary, the results show that there are routes that are relatively independent of the specific day of the week and time slot. For example, the rule {Passeig del Born} to {Catedral} appears in all time slots and is identified as the most frequent route in the city independently of the weekday. Similarly, the rule {Llotja de Palma, Catedral, Plaça del Mercat} to {Passeig del Born} describes three places in the city where the pedestrians come from when they go to {Passeig del Born}.

Therefore, comparing the results of each table shows that there are repeated association routes that appear in all tables. This confirms that mobility in the city has stable patterns during the week that can be considered for urban mobility planning and may be identified using the proposed methodology.

**Table 5.** Results for the routes during the leisure-entertainment time slot.

| Rule | Support (%) | | | | Confidence (%) | | | | Lift | | | |
|---|---|---|---|---|---|---|---|---|---|---|---|---|
| | Tuesday | Friday | Saturday | Sunday | Tuesday | Friday | Saturday | Sunday | Tuesday | Friday | Saturday | Sunday |
| {Passeig del Born} to {Catedral} | 43.5 | **48.7** | **52.67** | **56.0** | 56.14 | 61.2 | 66.2 | 67.4 | 1.12 | 1.09 | 1.13 | 1.09 |
| {Passeig del Born} to {Plaça del Mercat} | **47.8** | 44.7 | 44.7 | 49.4 | 61.54 | 56.3 | 56.3 | 59.5 | 1.2 | 1.19 | 1.19 | 1.15 |
| {Catedral} to {Passeig del Born} | 43.5 | **48.7** | **52.67** | **56.0** | 86.7 | 87.2 | 90.0 | 90.0 | 1.12 | 1.09 | 1.13 | 1.09 |
| {Llotja de Palma, Catedral, Plaça del Mercat} to {Passeig del Born} | 11.9 | 12.76 | 14.65 | 17.7 | **99.2** | **99.67** | **99.39** | **99.6** | 1.27 | 1.25 | 1.25 | 1.2 |
| {Plaça Major} to {Plaça de Cort} | 11.6 | 11.8 | 16.7 | 16.5 | 40.9 | 45.24 | 48.47 | 55.3 | **2.07** | **2.22** | **1.84** | **2.15** |
| {Plaça de Santa Eulalia} to {Plaça de Cort} | NA | NA | 11.3 | 10.7 | NA | NA | 81.5 | 83.66 | NA | NA | **3.11** | **3.25** |
| {Plaça de Cort} to {Plaça de Santa Eulalia} | NA | NA | 11.3 | 10.7 | NA | NA | 43.12 | 42.05 | NA | NA | **3.11** | **3.25** |

**Table 6.** Results for the routes during the rest period.

| Rule | Support (%) | | | Confidence (%) | | | Lift | | |
|---|---|---|---|---|---|---|---|---|---|
| | Wednesday | Thursday | Saturday | Wednesday | Thursday | Saturday | Wednesday | Thursday | Saturday |
| {Passeig del Born} to {Catedral} | **36.6** | **39.0** | **46.8** | 52.0 | 55.0 | 63.2 | 1.12 | 1.14 | 1.14 |
| {Plaça del Mercat} to {Passeig del Born} | **36.2** | **34.6** | **33.98** | 88.9 | 89.7 | 94.0 | 1.26 | 1.26 | 1.27 |
| {Catedral, Plaça del Mercat} to {Passeig del Born} | 13.7 | 14.0 | 17.0 | **97.0** | **96.7** | **98.9** | 1.38 | 1.36 | 1.33 |
| {Plaça Major} to {Plaça de Cort} | NA | NA | 12.2 | NA | NA | 51.1 | NA | NA | **2.34** |
| {Plaça d'Espanya, Passeig del Born} to {Plaça del Mercat} | NA | 10.6 | 10.6 | NA | 60.7 | 56.0 | NA | **1.57** | **1.55** |

## 6. Conclusions

This work proposes and applies a set of methodologies that can be integrated to model urban mobility using technical data provided by a public Wi-Fi network that registers digital devices that move around its coverage area.

The study proposes three integrated methodologies that allow us to classify the devices, estimate the mobility modes, and identify the most likely routes that people use to move around the city. The proposed methodologies have several advantages over alternative approaches. First, they do not require prior calibration conditions, although if these existed, they could be incorporated to improve the quality of the results. Second, the data cover a vast area of the city and a long period of analysis. Therefore, researchers can go beyond controlled spaces or limited time observational techniques. Additionally, the proposed methodologies do not require any participation from the city's users and respect all privacy regulations.

However, the application to open heterogeneous spaces, and to all multimodal movements, implies an initial phase of data refinement. This process is important since it eliminates outliers and optimizes the response time of the processing algorithms. Regarding the devices' classification, our empirical analysis found that during the application to Palma City, most devices were labeled as sporadic. However, it must be considered that we only evaluated one week's worth of data.

Our mobility mode classification concluded that pedestrian mobility was the most captured by the network, followed by the category "fast mode." This is an expected result as the network covers the city center with many pedestrian areas. This category could increase as citizens adopt sustainable transport vehicles (bicycles, scooters, etc.). Monitoring this evolution with public Wi-Fi networks facilitates progress in transport management policies in a smart city.

Finally, we identified the most common routes that people use to move around the city through the association rules' methodology and its corresponding reliability indicators. This methodology allows us to determine the combinations of public spaces (origin-destination) that people prefer. In addition, we present our analysis disaggregated by time frames to segment the most likely uses of the city and evaluate the variability in the estimated routes as a function of time.

The proposed methodology can be adapted to multiple temporal scenarios and space specifications of each city.

Hence, the main contributions of the paper are that: it combines both GIS information about singular places in the city and passive high-frequency Wi-Fi device detection; three techniques are described and applied over a week of device observations: device occurrence frequency, transport mode estimation, and the most common travel routes extraction. All techniques have been applied, considering their liability to be executed on time; therefore, their results can be obtained in near real time. Starting from raw technical data and without additional information about the devices, a methodology is proposed to differentiate between habitual and sporadic users, enabling the analysis to be extended to the field of tourist destination management. Moreover, it is possible to extend the analysis to other travel needs by just considering a different set of singular city places; Finally, the proposed methodology has been applied using available public Wi-Fi data, combining high time frequency and spatial precision to provide an accurate characterization of how the analyzed areas of a city are used with different time frequencies and geographical spread, showing its application in Palma.

This proposal uses raw technical data to obtain rich and dynamic urban mobility knowledge that can be used to improve the management and planning of infrastructures in a smart city context. In fact, improving urban mobility systems towards sustainability is one of the main goals of an eminently urban future. Moreover, monitoring urban mobility in real time has relevant applications for urban management and planning. Among them, it might be a relevant tool to identify episodes of urban overcrowding, which might be

particularly relevant under pandemic situations as which require monitoring social distance in real time.

As any other research, this study presents some limitations that open avenues for suture research. Specifically, the device location obtained using passive Wi-Fi data is only related to devices with an active Wi-Fi transmitter. The smartphone must have the Wi-Fi interface turned on; the route analysis is only related to areas with enough Wi-Fi coverage, so it is necessary to guarantee good Wi-Fi coverage in the city area. Moreover, due to the period difference in device detection, it is complex to determine the transport mode with more granularity. Of course, with the installation of a user app, more detailed modal analysis can be developed.

**Author Contributions:** Conceptualization, P.S., B.A.-L., M.R.-P. and V.R.; methodology, P.S., B.A.-L. and M.R.-P.; software, P.S., B.A.-L.; validation, P.S., B.A.-L., M.R.-P. and V.R.; formal analysis, P.S., B.A.-L., M.R.-P. and V.R; investigation, P.S., B.A.-L., M.R.-P. and V.R; resources, P.S., M.R.-P.; data curation, P.S., B.A.-L.; writing—original draft preparation, P.S.; writing—review and editing, P.S., B.A.-L., M.R.-P. and V.R.; visualization, P.S., M.R.-P.; supervision, P.S., B.A.-L., M.R.-P. and V.R.; project administration, P.S., M.R.-P.; funding acquisition, B.A.-L. and M.R.-P. All authors have read and agreed to the published version of the manuscript.

**Funding:** This work was sponsored by the *Comunitat Autonòma de les Illes Balears* through the *Direcció General de Política Universitària i Recerca* with funds from the Tourist Stay Tax Law (PRD2018/52– ITS 2017-006).

**Data Availability Statement:** Not applicable.

**Acknowledgments:** The authors acknowledge the collaboration of the Fundacio Turisme 365 Palma and of WIONGO, the firm responsible for the SmartWifi Palma network. Special thanks to Mauricio Socias, CEO of Wiongo for his support to the project and to Bartomeu Crespí for his support from the SmartOffice of the City Council of Palma.

**Conflicts of Interest:** The authors declare no conflict of interest. The funders had no role in the design of the study; in the collection, analyses, or interpretation of data; in the writing of the manuscript; or in the decision to publish the results.

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
