# Peer review of "Methodological Proposal for the Analysis of Urban Mobility Using Wi-Fi Data and Artificial Intelligence Techniques: The Case of Palma"

_electronics, doi:10.3390/electronics12030504_

Round 1

Reviewer 1 Report

Dear Authors, 

Thank you for providing this great review opportunity. 

This manuscript, titled 'Methodological proposal for the analysis of urban mobility using Wi-Fi data and artificial intelligence techniques: the case of Palma,' depicts relevant signals about urban mobility in tourist destinations like Palma de Mallorca. 

I have no reservations about using a method like random forest, but I believe poisson regression can also support your premise, particularly when studying frequency difficulties. This mobility pattern, I suppose, is influenced by the specific time periods and peak season of the trip to Palma. Please explain the context of the time and season chosen in this study project. 

Minor revisions would be suggested by analyzing feasible solutions that can supply each conceivable mobility route depending on their traveling needs (commercial, business, tourism, public work, etc.). This research should consider the future vision for quarantine density capacity following the COVID19 pandemic. 

Please also update the figures with better resolution versions. 

I would recommend that this work be accepted with minor revisions.

Author Response

Dear Editors and reviewers, 

Thanks for the time devoted to the improvement of our manuscript, for the positive comments, and for the opportunity to submit a revised manuscript. 

We have included all the previous comments in the new version of the manuscript. All new text is highlighted in blue color to facilitate the revision. 

Regarding the specific suggestions: 

I have no reservations about using a method like random forest, but I believe poisson regression can also support your premise, particularly when studying frequency difficulties. This mobility pattern, I suppose, is influenced by the specific time periods and peak season of the trip to Palma. Please explain the context of the time and season chosen in this study project 

Thanks for this comment. Yes, we do agree that poisson regression might have been an alternative approach.  

Regarding the mobility pattern, the data consider the high tourism season. And yes, as the reviewer mentions, it is likely to affect the mobility patterns, particularly in the case of tourists. 

As suggested, we have added the context of the time and season. The text is located just before Figure 1.  

Minor revisions would be suggested by analyzing feasible solutions that can supply each conceivable mobility route depending on their traveling needs (commercial, business, tourism, public work, etc.) 

Thank again for your suggestion. In the work, several city points are selected according to tourism interest. But also, other traveling needs can be included as the reviewer mention. We can work on that new needs in order to increase the mobility route impact. 

This research should consider the future vision for quarantine density capacity following the COVID19 pandemic 

Thanks for this Comment. Following this idea, we have added some sentences at the end of the manuscript considering the role of monitoring social distance 

Please also update the figures with better resolution versions 

Following this suggestion, we have updated the figures with higher resolution images  

Reviewer 2 Report

This study proposes the application of several Artificial Intelligence methodologies to sup- 15 port mobility planning based on data provided by public Wi-Fi infrastructures in the city. 

The paper is interesting however there is limited analysis of the existing literature. It is suggested to completely rewrite the paragraph on literature analysis.

Results need to be better contextualized with reference to existing literature

What are the strengths and weaknesses of the work? this needs to be clarified

Minor:

keywords are missing

I wish the authors good continuation with this work

Author Response

Dear Editors and reviewers, 

Thanks for the time devoted to the improvement of our manuscript, for the positive comments, and for the opportunity to submit a revised manuscript. 

We have included all the previous comments in the new version of the manuscript. All new text is highlighted in blue color to facilitate the revision. 

Regarding the specific suggestions: 

The paper is interesting however there is limited analysis of the existing literature. It is suggested to completely rewrite the paragraph on literature analysis. Results need to be better contextualized with reference to existing literature 

Following the suggestion, we have extended the literature revision to contextualize our research. 

What are the strengths and weaknesses of the work? this needs to be clarified  

Thanks for this question. Following that suggestion, we have added several paragraphs at the end of the conclusions emphasizing the main strengths and limitations of the paper 

keywords are missing

We have added the keywords. They can be found just after the abstract 

Round 2

Reviewer 2 Report

the indications were well received by the authors